# Ranking Architectures by Feature Extraction Capabilities

**Debadeepta Dey, Shital Shah, Sebastien Bubeck** *Microsoft Research*
DEDEY,SHITALS,SEBUBECK@MICROSOFT.COM

## Abstract

The fundamental problem in Neural Architecture Search (NAS) is to efficiently find high-performing ones from a search space of architectures. We propose a simple but powerful method for ranking architectures `FEAR` in any search space. `FEAR` leverages the viewpoint that neural networks are powerful non-linear feature extractors. By training different architectures in the search space to the *same* training or validation error and subsequently comparing the usefulness of the features extracted on the task-dataset of interest by freezing most of the architecture we obtain quick estimates of the relative performance. We validate `FEAR` on Natsbench topology search space on three different datasets against competing baselines and show strong ranking correlation especially compared to recently proposed zero-cost methods. `FEAR` especially excels at ranking high-performance architectures in the search space. When used in the inner loop of discrete search algorithms like random search, `FEAR` can cut down the search time by $\approx 2.4X$ without losing accuracy. We additionally empirically study very recently proposed zero-cost measures for ranking and find that they breakdown in ranking performance as training proceeds and also that data-agnostic ranking scores which ignore the dataset do not generalize across dissimilar datasets.

## 1. Introduction

We propose a method to speed-up the evaluation phase of discrete NAS methods (10; 23) where it is often an ad-hoc choice on how long to evaluate each sampled architecture for a given dataset and is the most expensive part of such methods (34; 36). Specifically we propose a simple but powerful architecture ranking methodology that enables good ranking by leveraging the fact that neural networks are powerful feature extractors and the power of an architecture is dependent on how effective it is at extracting useful features for the given task. We term this 'recipe' (see Figure 1 for an overview) `FEAR` (FEATure-extraction Ranking) which we discuss at length in Section 3.

We have two main contributions: 1. A simple fast architecture evaluation method named `FEAR` and validate it on a variety of datasets on the Natsbench topological space benchmark against competing baselines.[1] 2. We also empirically find that a number of very recently proposed lightweight ranking measures (1; 19) degrade in ranking performance as network training progresses and that data-agnostic ranking measures especially don't generalize across datasets. The performance of an architecture is a function of *both the topology and the dataset* (in addition to the training pipeline).

## 2. Related Work

Here we discuss the works that are directly relevant to fast evaluation and ranking of architectures.

---

1. Reproducible implementation of all experiments are available at `https://github.com/microsoft/archai/tree/fear_ranking` (25).

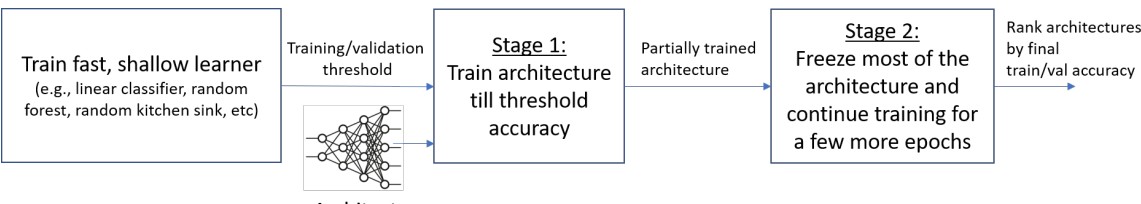

Figure 1: Overview of `FEAR` which first trains a fast but shallow learner to get a reasonable training/validation error threshold and then trains the architecture in a two-stage procedure. In the first stage the architecture is regularly trained until it achieves threshold accuracy. In the second stage most of the partially trained architecture from the first stage is frozen and training continues for a few more epochs. All candidate architectures can then be ranked by final training or validation accuracy obtained via this two stage procedure.

**Architecture Performance Prediction:** Since this body of work is orthogonally related but important to discuss we detail it in Appendix A.2.

**Lightweight Architecture Evaluation:** Zhou et al. (37) (EcoNAS) search for combinations of reduced input image resolution, fewer epochs, and number of stem channels to find computationally cheap proxies for evaluating architectures while keeping their relative ranks the same. They find an optimal configuration of resolution, epochs and number of channels on a bag of 50 models. Abdelfattah et al. (1) note that the configuration found by EcoNAS suffers from degrading performance when evaluated on all 15625 models in Nasbench-201 CIFAR10 dataset (8). Abdelfattah et al. (1) conduct their own search and find a different configuration that works better on Nasbench201 CIFAR10 dataset. They caution that such proxies clearly don't work on different search spaces even when the dataset and task are the same and also the importance of measuring actual wallclock run times as reduced flops often don't translate into actual time savings due to different ways of accessing memory.

**Trainingless Proxies:** Mellor et al. (19) propose a *trainingless* method for ranking architectures based on the KL divergence between an uncorrelated Gaussian distribution and the correlation matrix of local linear operators (for networks with ReLU activation functions) associated with every data point. If the correlation between such maps is low then the network should be able to model each data point well. Since this score can be computed with just a small sample of the dataset (a single minibatch), it takes negligible compute and time. In recent work Abdelfattah et al. (1) thoroughly empirically evaluate this method (termed as `jacob_cov` ) along with an entire family of pruning-at-initialization schemes which they convert to trainingless ranking methods by simply summing up the saliency scores at each weight. The particular methods include `snip` (15), `grasp` (32), `synflow` (28) and `fisher` (31). On Nasbench-201, on all three datasets (CIFAR10, CIFAR100, ImageNet16-120) they find that `synflow` and `jacob_cov` scores performed very well. A majority vote amongst `synflow` , `jacob_cov` and `snip` termed as `vote` performs the best.

   Note that `FEAR` is *not* a trainingless method and does use more computation than trainingless proxies. But we empirically show in Section 4 that `FEAR` outperforms these proxy measures as well as the natural baselines of reduced number of training epochs. In the process of experimentation we identify some curious properties of trainingless proxies like degradation in performance as the network trains more which is counter-intuitive and also the curious phenomenon of `synflow` -based ranking (which is a data-agnostic scoring

mechanism) in particular not generalizing across datasets (See Appendix A.4). This supports the intuition that architecture performance is not an intrinsic property of just its topology (and training procedure) but crucially also dependent on the task and dataset at hand. This has been empirically validated by the very recent work of Tuggener et al. (30) who show that architectures which perform well on ImageNet (7) do not necessarily perform as well on other datasets. On some datasets their ranks are negatively correlated with ImageNet ranks. This further suggests that a data-agnostic scoring mechanism like `synflow` should not work well at ranking architectures.

## 3. Approach

**Training accuracy threshold:** Figure 1 shows a high level schematic of the approach. `FEAR` first trains a fast but shallow learner on the dataset and task of choice to learn a training or validation accuracy threshold. Specifically, for the task of image classification one can use a number of fast shallow learners like random forest, linear classifier, with handcrafted visual features like Histogram-of-Gradients (HoG) (6) or random features like random kitchen sink (22). We emphasize that the role of this threshold is to be both non-trivial yet not too difficult to beat with a neural network architecture.

**Stage 1: Regular training till threshold accuracy:** `FEAR` trains the candidate architecture till it achieves this threshold accuracy.

**Stage 2: Using architecture as feature extractor:** `FEAR` then freezes most of the layers of the architecture other than the last few layers and trains for a few more epochs. *Freezing* is several times faster per step as gradients are not computed for most layers. This stage treats the network as a feature extractor and trains a relatively shallow network utilizing these features. A pool of candidate architectures are then ranked by their final training or validation accuracies on the dataset under consideration. Intuitively `FEAR` ranks architectures on their ability to extract useful features from inputs.

A question that arises is by cutting off training of most of the layers at a relatively early stage of training, are we not hurting the architecture's ability to potentially distinguish itself at feature extraction? Raghu et al. (21) and Kornblith et al. (13) dive deep into the training dynamics of neural networks and show that networks train 'bottom-up' where the bottom layers (near the input) train quite fast early-on in training and become stable. As training progresses bottom layers rarely change their representation and mostly the top layers change to learn decision-making rules using the bottom layers as rich feature extractors. `FEAR` leverages this insight.

**Role of training till threshold accuracy:** We re-emphasize that `FEAR` does not fix the number of epochs apriori. This has a number of advantages. First, it makes architectures *comparable* to each other and makes sure that every architecture gets ample time to learn the best features it can for the task. Secondly, it exposes first-order dynamics of training in that weak architectures take longer time to reach the same training or validation error compared to stronger architectures. See Figure 2 where we plot for CIFAR10 the time taken by 1000 architectures to reach a threshold accuracy (x-axis) against the final test error (y-axis). (Appendix A.5 for all three datasets.) Invariably architectures which go on to attain good final test accuracy achieve threshold training accuracy much faster. Note that there are no architectures that train slowly but go on to achieve good final test accuracy after full

training ('late-bloomers', these would have been on the upper-right hand part of the plots). This effect can be utilized to early-stop evaluation of candidates that take much longer than the fastest architecture encountered to reach threshold accuracy. This is in fact crucial to obtain large speedup when combining our ranking method with standard discrete neural architecture search techniques.

**Motivation:**   The motivation for our method comes from the emerging theoretical understanding of gradient learning on neural networks. It has been observed (e.g., (11; 4; 20; 2)) that initially training happens in the so-called "neural tangent kernel" (NTK) (12) regime, where the network basically uses its initialization as a sort of kernel embedding and performs a kernel regression. Our key hypothesis is that this phase should stop a bit before reaching the threshold accuracy, since this threshold has been obtained with a good kernel method (or something slightly more powerful like a random forest model). In other words, when we stop the training after reaching the threshold accuracy, it should be that the network has already escaped the NTK regime and is currently actively training the features (second phase of learning). Our second (empirical) hypothesis is that the quality of the features learned in the early part of this second phase is predictive of the final quality of the network. We measure the quality of the learned features via the freezing technique, whose extreme case is to only continue training the final layer (i.e., a linear model on top of the current embedding).

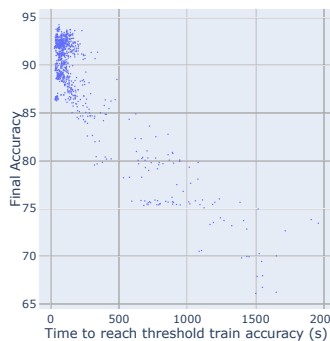

Figure 2: Time to reach threshold training accuracy (x-axis) vs. final test accuracy on 1000 uniformly sampled architectures on Natsbench CIFAR10. Worse-performing architectures take longer time to reach threshold training accuracy than stronger ones.

## 4. Experiments

**Search space:**   We uniformly randomly sample 1000 architectures from the 15625 architectures of the topology search space of the Natsbench (9) benchmark and hold them constant for all following experiments. The topology search space is similar to that used in DARTS (18). See Figure 1 in (9) for a visualization of the search space. Natsbench topology search space has trained each architecture on CIFAR10, CIFAR100 (14) and ImageNet16-120 (5) image classification datasets.

**Spearman's rank correlation vs. evaluation wall-clock time:**   We report performance of `FEAR` and baselines by first binning architectures into several buckets of increasing size. For example `Top` 10% in Figure 5 shows the average wall-clock time taken by any method (x-axis) vs. Spearman's rank correlation (27)[2] of the method with the groundtruth rank of architectures (by test accuracy) after full final training procedure over the top 10% of architectures. Similarly the bin of `Top` 20% architectures includes top 20% of candidates and so on. We break-up the performance of methods over such cumulative bins to highlight

---

2. Spearman's rank correlation is between $(-1, 1)$ with 1 implying perfect correlation and $-1$ anti-correlation of the *ranks* of candidate architectures.

how methods perform in discriminating amongst high-performing candidates and not just over the entire population, since it is crucial for any reduced-computation proxy ranking method to hone-in on good ones especially and not just the entire population.

**Percentage overlap with groundtruth ranking vs. evaluation wall-clock time:** It is also important to evaluate what percentage of architectures are common between the top $x\%$ of groundtruth architectures as ranked by some method. This evaluates when asked to rank the entire population, do high ranking ones in groundtruth end up in high ranking places (and vice-versa). By definition this is between $(0, 1)$.

**Regular training with reduced epochs: `shortreg`** The natural baseline is to compare `FEAR` against reduced epochs of training. Most NAS methods use a reduced number of training epochs (16) in the inner loop to decide the relative ranks of architectures as a proxy for final performance after undergoing the complete training procedure. We term this as `shortreg`. We show that `FEAR` consistently outperforms the pareto-frontier of wall-clock time vs. Spearman rank correlation and ratio of common architectures at cumulative bins of candidate architectures by groundtruth (test accuracy).

**Zero-cost Proxies:** We evaluate zero-cost proxies (1) for ranking and observe a number of mysterious phenomena where such measures break down across datasets and as networks are trained. This shows that such measures while exhibiting reasonable prima-facie performance on NAS benchmarks are not generalizing across datasets. We detail our investigation in Appendix A.4.

**Reduced resolution Proxies:** While reduced resolution proxies as proposed in (37) and (1) can significantly speed up architecture evaluation, note that they are orthogonal to our approach as they equally speed-up both `shortreg` and `FEAR` .

**Training procedure hyperparameters and hardware:** All experiments use the same hyperparameter settings as Natsbench, specifically cosine learning rate schedule with starting rate of 0.1, minimum rate of 0.0, SGD optimizer, decay of 0.0005, momentum 0.9 and Nesterov enabled. For `shortreg` baseline we vary number of epochs and batch sizes $256, 512, 1024, 2048$ to find a pareto-frontier of wall-clock time vs. Spearman's correlation and common ratio. We especially investigate varying the batch size since that can have a large effect on wall-clock time. All experiments were conducted on Nvidia V100 16 GB GPUs.

### 4.1 Main Results

We show the pareto-frontier of performance between `FEAR` and `shortreg` in Appendix A.6 (figures 5, 6 and 7) on Natsbench CIFAR10, CIFAR100 and ImageNet16-120 respectively. We order architectures in descending order of test accuracy, and bin them cumulatively into top 10%, 20%, ... bins. For each bin we report the two performance criteria detailed above. For better elucidation, in Table 1 for each bin we note `FEAR` and the nearest point on the pareto-frontier generated by `shortreg` using Spearman's correlation ('spe') and common ratio of architectures in groundtruth ranking against the average time in seconds. Especially on CIFAR10 and CIFAR100 large gaps in performance can be seen at high-performing architecture bins. In Table 3 in Appendix A.6 we note the corresponding tables for CIFAR100 and ImageNet16-120.

| Top % | FEAR (spe, s) | Nearest Pareto (spe, s) | FEAR (common, s) | Nearest Pareto (common, s) |
|---|---|---|---|---|
| 10 | **0.22**, **133.22**$^{\pm 4.09}$ | 0.16, 138.28$^{\pm 2.25}$ | **0.59**, **133.22**$^{\pm 4.09}$ | 0.40, 138.28$^{\pm 2.25}$ |
| 20 | **0.40**, **133.04**$^{\pm 2.76}$ | 0.19, 134.68$^{\pm 1.50}$ | **0.60**, **133.04**$^{\pm 2.76}$ | 0.57, 134.68$^{\pm 1.50}$ |
| 30 | **0.41**, **132.76**$^{\pm 2.24}$ | 0.28, 140.13$^{\pm 1.46}$ | **0.70**, **132.76**$^{\pm 2.24}$ | 0.70, 132.77$^{\pm 2.24}$ |
| 40 | **0.43**, **131.41**$^{\pm 2.00}$ | 0.34, 135.31$^{\pm 1.32}$ | **0.78**, **131.41**$^{\pm 2.00}$ | 0.72, 135.31$^{\pm 1.32}$ |
| 50 | **0.55**, **134.48**$^{\pm 1.97}$ | 0.50, 136.26$^{\pm 1.14}$ | **0.79**, **134.48**$^{\pm 1.97}$ | 0.79, 136.26$^{\pm 1.14}$ |
| 100 | 0.83, 236.36$^{\pm 9.39}$ | **0.90**, **218.97**$^{\pm 1.44}$ | 1.0, 236.36$^{\pm 9.39}$ | **1.00**, **218.97**$^{\pm 1.44}$ |

Table 1: (Left) Spearman's correlation ('spe') comparison between FEAR and the nearest point on the pareto-frontier etched out by shortreg variants with respect to average wall-clock time (s). (Right) Ratio of overlap in bins between rankings of FEAR and nearest pareto-frontier point of shortreg . The nearest pareto-frontier points are found by inspecting figure 5

For CIFAR10, CIFAR100 FEAR consistently places above the pareto-frontier of shortreg especially at higher ranked architectures. FEAR is able to both discriminate better amongst high-performing architectures with shorter wall-clock time and achieve ranking which overlaps more with the groundtruth. As the bin increases to encompass the entire set of 1000 architectures (Top 100%) by construction FEAR takes more time as low-performing architectures take longer to reach threshold accuracy (recall Figure 4). In practice, this extra time for lower-performing architectures will not be paid since low-performing ones can be simply removed from consideration when they exceed the fastest time to achieve threshold accuracy till then. On ImageNet16-120 the gap between FEAR and shortreg is not as significant but nevertheless it doesn't degrade in performance below shortreg and over the high-performance bins is marginally better.

**Deeper dive into zero-cost measures**   In Appendix A.4 we detail experiments on zero-cost measures. Our main findings are that counter-intuitively many zero-cost measures degrade drastically in performance as training progresses. Also that on a synthetic dataset zero-cost measures have almost no ranking correlation while FEAR still performs reasonably.

**Random Search with FEAR:**   On CIFAR100 we ran 10 trials random search (RS) with FEAR with different random seeds where the search cut-off any architecture which exceeded 4.0 times the fastest time to reach threshold training accuracy (0.3) encountered so far. This obtained a final accuracy of $72.04 \pm 0.29\%$ with $142550 \pm 3106$ seconds search time. Random search with shortreg of 50 epochs obtained $72.08 \pm 0.30\%$ with $347639 \pm 1674$ seconds search time. RS-FEAR can get same final accuracy within experimental error while being $\approx 2.43$ times faster. Each method got a budget of 500 architectures and the same set of 10 random seeds so that each method encountered the same sequence of architectures.

**Ongoing Work**   We are validating FEAR on currently state-of-the-art discrete search methods (34) or even simpler techniques which have been surprisingly effective on NAS benchmarks

like (36), larger search spaces like DARTS via Nasbench-301 (26) and search spaces around Transformer-like architectures (33; 29).

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

## Appendix A. Appendix

### A.1 Reproducibility and Best Practices Checklist in NAS

We use the best practices by (17) to foster reproducibility and do better empirical NAS research.

1. Best practices for releasing code.

   (a) Code for the training pipeline used to evaluate the final architectures. - Yes.
   (b) Code for the search space. - Yes.
   (c) The hyperparameters used for the final evaluation pipeline as well as random seeds. - Yes.
   (d) Code for your NAS method. - Yes.
   (e) Hyperparameters for your NAS method, as well as random seeds. - Yes.

2. Best practices for comparing NAS methods.

   (a) For all NAS methods you compare, did you use exactly the same NAS benchmark, including the same dataset (with the same training-test split), search space and code for training the architectures and hyperparameters for that code? - Yes.
   (b) Did you control for confounding factors (different hardware, versions of DL libraries, different runtimes for the different methods)? - Yes. Specifically we run all the baselines ourselves on the same hardware.
   (c) Did you run ablation studies? - Yes.
   (d) Did you use the same evaluation protocol for the methods being compared? - Yes.
   (e) Did you compare performance over time? - Yes.
   (f) Did you compare to random search? - Yes.
   (g) Did you perform multiple runs of your experiments and report seeds? - Yes.
   (h) Did you use tabular or surrogate benchmarks for in-depth evaluations? - Yes.

3. Best practices for reporting important details. - Yes

   (a) Did you report how you tuned hyperparameters, and what time and resources this required? - Yes.
   (b) Did you report the time for the entire end-to-end NAS method (rather than, e.g., only for the search phase)? - Yes.
   (c) Did you report all the details of your experimental setup? - Yes.

### A.2 Orthogonally Related Work

**Architecture Performance Prediction:** Baker et al. (3) propose training regressors which take in the architecture, training hyperparameters and the first few validation accuracies as features and try to predict the final validation accuracy. The proposed method has a "burn-in" phase where a sampled few architectures are first fully trained to gather data for training the regressor. This regressor is then utilized in the rest of the pipeline. Similarly

Rorabaugh et al. (24) propose fitting curves to the performance numbers of the first few training iterations and extrapolate to later accuracy values. White et al. (34) study neural network performance prediction in the context of Bayesian Optimization. Orthogonally White et al. (35) conduct an extensive study of architecture encodings for common NAS algorithm subroutines including performance prediction which sheds light on the pros and cons of certain featurizations for this task. `FEAR` is orthogonal to the above body of work on architecture performance prediction and in fact can be used for further speeding up the performance prediction modules as one doesn't have to train the architectures fully to create the training set.

## A.3 Finding the training threshold for a dataset

We construct a shallow pipeline using Histogram-of-Oriented-Gradients (HoG) (6) as image features and construct a relatively shallow learner by passing the features through two hidden fully connected layers. This simple pipeline achieves 0.6 training accuracy on CIFAR10, 0.3 on CIFAR100 and 0.2 accuracy on ImageNet16-120. These numbers were used as the training accuracy threshold for stage 1 of `FEAR` with respective datasets.

## A.4 Deeper Dive into Zero-Cost Measures

As discussed in Section 2, Abdelfattah et al. (1) propose using pruning-at-initialization methods like `synflow` , `snip` , `grasp` , `fisher` etc for ranking architectures without any training by summing up the per-weight saliency scores to come up with an overall architecture score. In thorough experiments, `synflow` emerged as a really good ranking measure with a majority voting scheme with `synflow` , `jacob_cov` and `snip` emerging as the best overall. As discussed in Section 2, `synflow` 's good performance is a bit perplexing since it is a data-agnostic measure. It suggests that there are inherently good and bad architectures and the particulars of the dataset should not matter. In order to investigate this we created a synthetic dataset with properties such that it would be a drop-in for CIFAR10. Specifically we created random Gaussian images of dimension `[32, 32, 3]`. Each image was assigned a class label in $(0, 10)$ by passing each image through 10 randomly initialized neural networks and picking the id of the network which assigned the image a maximum score. Each of the networks has a simple architecture of a linear layer with dimension 3072, followed by a `ReLu` layer, followed by a linear layer which produces a single scalar output. A dataset of 60000 images was generated with 50000 training and the rest held-out as a test set. Each class has 6000 examples. We refer to this dataset asSynthetic CIFAR10 .

The same set of 1000 randomly sampled architectures were evaluated on this dataset using `FEAR` and the various zero-cost measures. Table 2 shows that the various zero-cost measures have almost no correlation with rankings while `FEAR` still works reasonably. Also note that the rankings via `synflow` which ignore the dataset are no longer valid on Synthetic CIFAR10 . This means that architectures which performed really well on CIFAR10 don't work as well on Synthetic CIFAR10 . This is at least an existence proof of the fact that performance of an architecture is also a function of the dataset and task and is not an inherent property of just the topology of the network as also empirically shown recently by Tuggener et al. (30).

In Figure 3 we evaluated zero-cost measures after each epoch of training on CIFAR10 for the 1000 randomly sampled architectures from Natsbench topological search space. We find

| Method | Spearman Corr. |
|---|---|
| synflow | −0.000437 |
| jacob_cov | −0.13 |
| snip | −0.31 |
| fisher | −0.42 |
| grasp | −0.25 |
| synflow_bn | 0.18 |
| FEAR | 0.55 |

Table 2: Zero-Cost measures on Synthetic CIFAR10 shows achieve low correlation while FEAR still gets reasonable performance. This phenomenon is especially not amenable for data-agnostic measures like synflow which ignore the dataset completely.

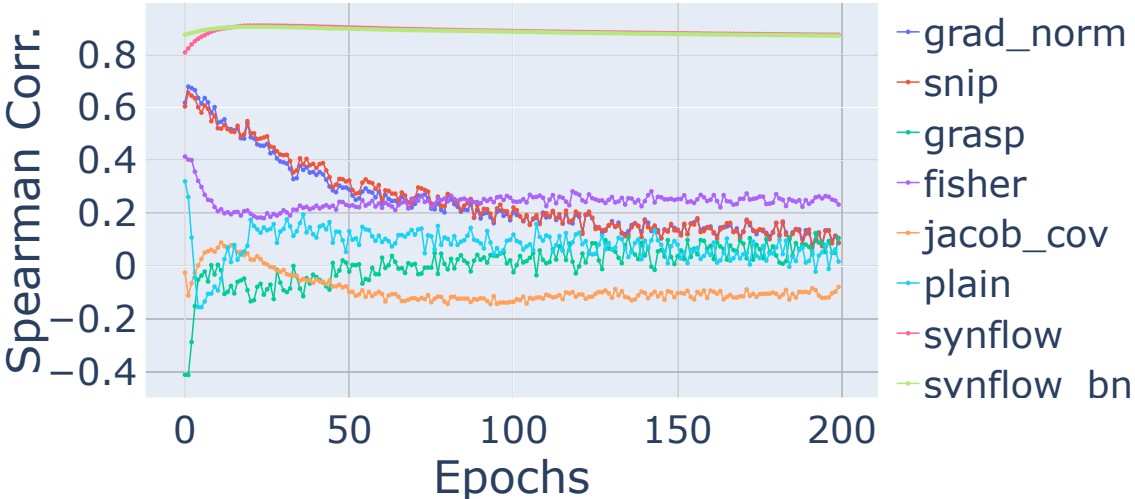

Figure 3: Evaluating the ranking performance of zero-cost measures after each epoch of training. Measures like grasp and jacob_cov demonstrate large degradation in performance after even a single epoch of training. snip and grad_norm decay gradually.

that measures like snip and grad_norm gradually degrade in rank correlation as the network trains. jacob_cov and grasp at initialization have Spearman of 0.69 and 0.63 respectively but after even one epoch of training drastically degrade to −0.02 and −0.41. Note from figures 5, 6, and 7 that ranking architectures via training error even after one or two epochs of training leads to much better ranking correlation.

Note that measures derived from snip , grasp , synflow are intended originally for the task of *pruning* architectures at initialization. So it is perhaps not surprising that the saliency scores when summed-up over individual weights to provide a global architecture score doesn't exhibit good ranking performance as the network is trained.

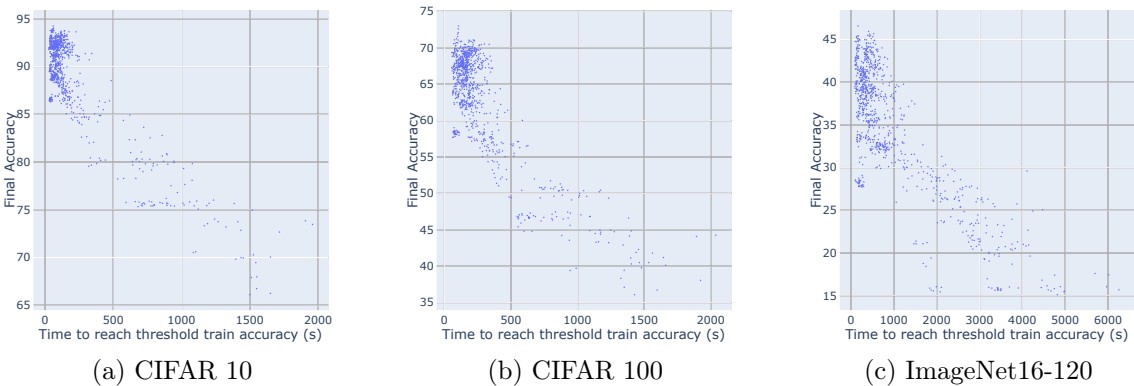

(a) CIFAR 10     (b) CIFAR 100     (c) ImageNet16-120

Figure 4: Time to reach threshold training accuracy (x-axis) vs. final test accuracy on 1000 uniformly sampled architectures on Natsbench. They show a clear relationship where ultimately worse performing architectures take longer time to reach threshold accuracy than stronger ones.

## A.5  Test Duration vs. Final Test Accuracy

## A.6  Detailed Ranking Plots on Natsbench Topological Search Space

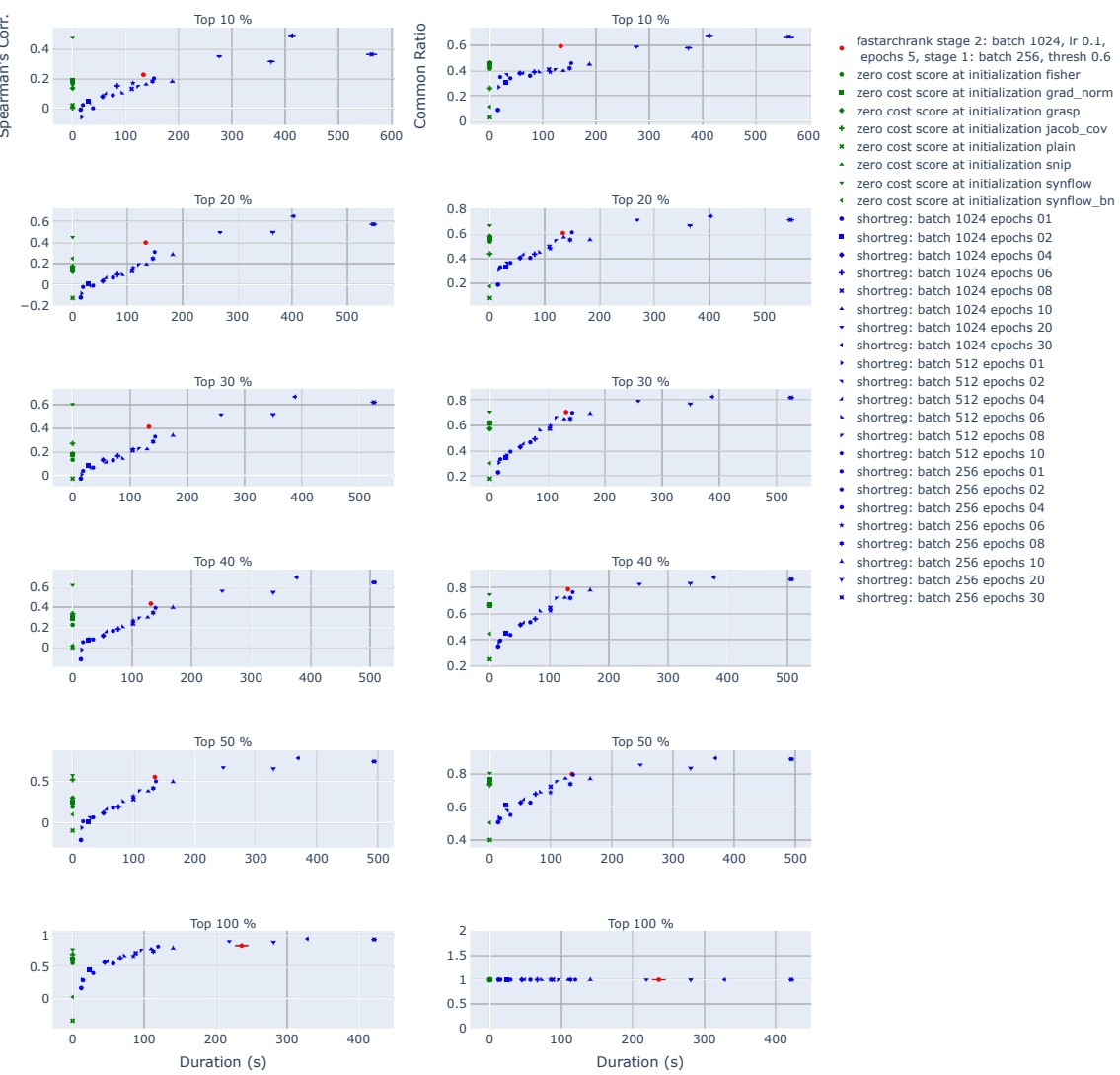

Figure 5: [Left] Average duration vs. Spearman's correlation and [Right] average duration vs. common ratio over the top $x\%$ of the 1000 architectures sampled from Natsbench topological search space on CIFAR10. We also show the various zero-cost measures from Abdelfattah et al. (1) in green.

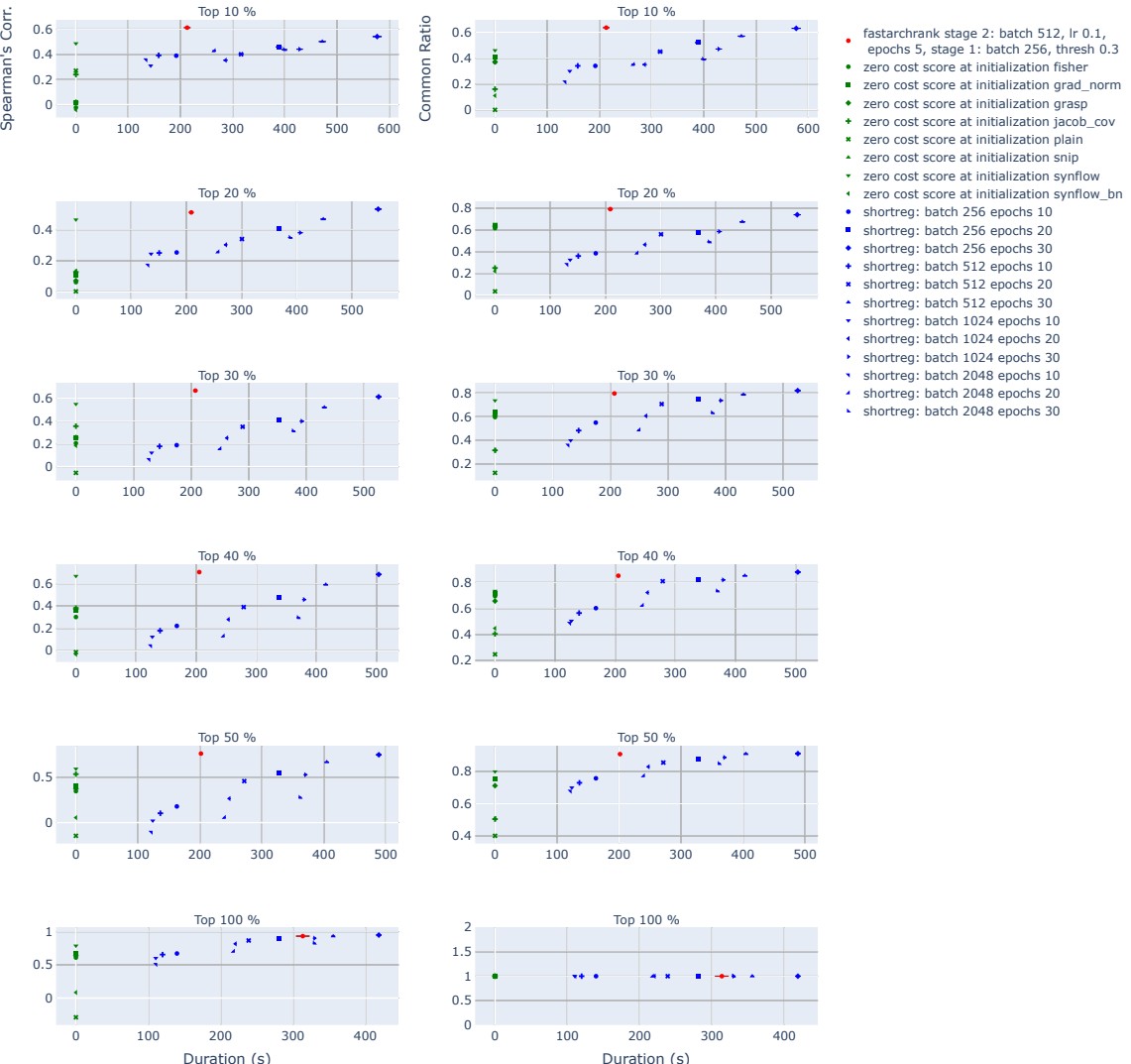

Figure 6: [Left] Average duration vs. Spearman's correlation and [Right] average duration vs. common ratio over the top $x\%$ of the 1000 architectures sampled from Natsbench topological search space on CIFAR100. We also show the various zero-cost measures from Abdelfattah et al. (1) in green.

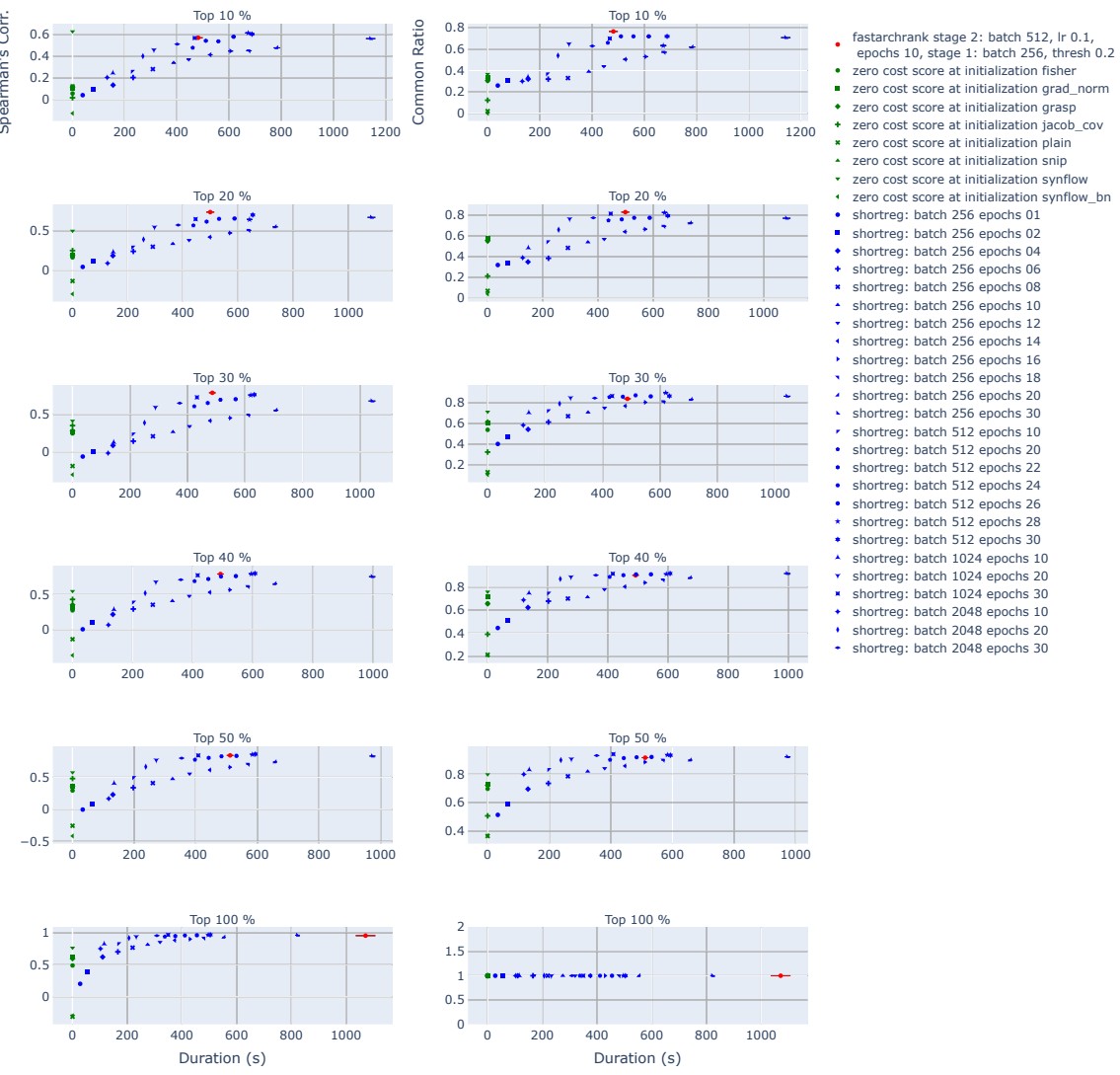

Figure 7: [Left] Average duration vs. Spearman's correlation and [Right] average duration vs. common ratio over the top $x\%$ of the 1000 architectures sampled from Natsbench topological search space on ImageNet16-120. We also show the various zero-cost measures from Abdelfattah et al. (1) in green.

| Top % | FEAR (spe, s) | Nearest Pareto (spe, s) | FEAR (common, s) | Nearest Pareto (common, s) |
|---|---|---|---|---|
| 10 | **0.61**, **213.07$^{\pm 6.33}$** | 0.42, 264.68$^{\pm 3.52}$ | **0.63**, **213.04$^{\pm 6.33}$** | 0.63, 264.68$^{\pm 3.52}$ |
| 20 | **0.51**, **208.84$^{\pm 4.36}$** | 0.25, 257.12$^{\pm 2.43}$ | **0.79**, **208.84$^{\pm 4.36}$** | 0.79, 257.12$^{\pm 2.43}$ |
| 30 | **0.66**, **207.19$^{\pm 3.63}$** | 0.15, 249.86$^{\pm 2.03}$ | **0.79**, **207.19$^{\pm 3.63}$** | 0.79, 249.86$^{\pm 2.03}$ |
| 40 | **0.70**, **205.13$^{\pm 3.16}$** | 0.27, 253.67$^{\pm 2.18}$ | **0.85**, **205.13$^{\pm 3.17}$** | 0.85, 244.96$^{\pm 1.85}$ |
| 50 | **0.76**, **201.65$^{\pm 2.78}$** | 0.26, 247.37$^{\pm 1.97}$ | **0.90**, **201.65$^{\pm 2.78}$** | 0.90, 239.41$^{\pm 1.66}$ |
| 100 | **0.93**, **313.52$^{\pm 9.31}$** | 0.90, 329.79$^{\pm 2.19}$ | 1.0, 313.52$^{\pm 9.31}$ | **1.00**, **281.30$^{\pm 2.42}$** |

(a) Natsbench CIFAR 100

| Top % | FEAR (spe, s) | Nearest Pareto (spe, s) | FEAR (common, s) | Nearest Pareto (common, s) |
|---|---|---|---|---|
| 10 | 0.56, 481.75$^{\pm 17.14}$ | **0.56**, **468.17$^{\pm 6.07}$** | **0.76**, **481.75$^{\pm 17.14}$** | 0.72, 510.86$^{\pm 6.77}$ |
| 20 | **0.73**, **498.91$^{\pm 14.52}$** | 0.64, 530.86$^{\pm 5.98}$ | 0.82, 498.91$^{\pm 14.52}$ | **0.81**, **445.89$^{\pm 5.05}$** |
| 30 | **0.78**, **486.40$^{\pm 11.56}$** | 0.69, 514.65$^{\pm 4.99}$ | 0.83, 486.40$^{\pm 11.56}$ | **0.85**, **470.47$^{\pm 4.58}$** |
| 40 | **0.79**, **491.91$^{\pm 10.74}$** | 0.75, 493.66$^{\pm 4.61}$ | 0.90, 491.91$^{\pm 10.74}$ | 0.91, 493.66$^{\pm 4.61}$ |
| 50 | 0.84, 571.37$^{\pm 10.73}$ | **0.83**, **532.04$^{\pm 4.70}$** | 0.91, 511.37$^{\pm 10.73}$ | 0.88, 510.74$^{\pm 5.04}$ |
| 100 | 0.95, 1070.88$^{\pm 35.88}$ | **0.96**, **822.05$^{\pm 8.14}$** | 1.0, 1070.88$^{\pm 35.88}$ | **1.00**, **822.05$^{\pm 8.14}$** |

(b) Natsbench ImageNet16-120

Table 3: (Left) Spearman's correlation ('spe') comparison between FEAR and the nearest point on the pareto-frontier etched out by shortreg variants with respect to average wall-clock time (s). (Right) Ratio of overlap in bins between rankings of FEAR and nearest pareto-frontier point of shortreg . The nearest pareto-frontier points are found by inspecting figures 5, 6 and 7

