# OpenReview forum: "Ranking Architectures by Feature Extraction Capabilities"
_ICML.cc/2021/Workshop/AutoML — AutoML@ICML2021 Poster_

### Official Review · Reviewer_ZsoV · 2021-06-10
**Empirically-driven training-based NAS proxy perform (slightly) better than traditional reduced-training NAS proxies**

**Rating:** 6
**Confidence:** 5

**Review:**

** This paper wasn't a blind submission. I could see the author's names and affiliations. **

This paper presents a new reduced-training proxy for use within neural architecture search. The presented "FEAR" proxy differs from traditional reduced-training proxies (e.g. EcoNAS) in that most of the parameters are frozen for part of the training. Specifically, the whole DNN is trained to a specific accuracy that matches a shallow classifier, the parameters in the feature extractor are then frozen, then the remaining (last few) layers continue to be trained for a few epochs before final evaluation. The authors present some very interesting intuitions (and references to prior work) to motivate their methodology. However, I would say that the paper's contribution mainly hinges on the empirical evaluation of the new proxy.

A single NAS benchmark (subset of NATSBench) is used for evaluation with a thorough comparison to both zero-cost proxies and "shortreg" reduced-training proxies. The results show that FEAR can outperform shortreg (albeit with a small margin in many cases). A tangential analysis is done on zero-cost proxies, specifically, identifying failure cases on datasets such as synthetic-CIFAR.

Overall, I found the paper interesting to read, with many astute and informative observations on existing NAS proxies. On the negative side, I found the evaluation to be quite limited: 1 NAS benchmark, single-seed (I think?), no evaluation with NAS algorithms (just rank correlations), no discussion of FEAR shortcomings (for example large datasets e.g. ImageNet, JFT?). To me, this paper beats the acceptance threshold because the topic is interesting, new and relevant, and the paper itself is well written. However, I would comment that I would need to see more results before definitively saying that FEAR is a robust or powerful proxy for use within NAS.

---

### Official Review · Reviewer_cQde · 2021-06-17

**Rating:** 7
**Confidence:** 5

**Review:**

The authors introduce a simple yet powerful approach to evaluate the potential of neural architectures. First, a validation threshold is achieved by shallow learners, which is further used to stop the architecture's training. The architecture is trained for a few more epochs with frozen weights for a faster fine-tuning. In the end, architectures are ranked by the final validation accuracy and the time needed to reach the threshold.

Pros:
- Approach is written clearly and thus easy to understand.
- Figures support the work nicely.
- FEAR has the potential of becoming a good baseline as it is easy to implement, and the number of epochs is not needed apriori.

Negative:
- It is not clear which linear classifiers are used or how they were chosen.
- Results are only compared to a training with reduced epochs.
- Why are architectures trained after the threshold is achieved? The computation time is already a strong indicator if it comes to ranking only. A comparison with and without "Stage 2" would be surely interesting.
- It is not well explained how the layers are frozen in the end.

---

### Official Review · Reviewer_zbsq · 2021-06-18
**A simple method to rank architectures but has some flaws**

**Rating:** 4
**Confidence:** 4

**Review:**



-- Summary --
This work proposes a simple architecture ranking scheme for NAS, namely FEAR. FEAR first train the archtiectures to a threshold accuracy, then freeze most of the layers of the architectures and continue to train for a few more epochs. Then they use the final prediction to rank these architectures.

Experiments are done on NATS-Bench only.

-- Strength --
+ The proposed approach relies on the assumption that weaker archtiecture takes longer time to reach the threshold accuracy, which seems reasonable. Experiments of spearman correlation also reveals this fact.
+ Method is simple and probably easy to implement.

-- Weakness --
There are several weakness of this work.

1. Training the architecture for a short period of time and use the performance to rank architecture is developed in some previous work, such as Hu et al. Angle-based Search Space Shrinking for Neural Architecture Search. This undermines the novelty. Especially I did not see any empirical comparison.

2. This work samples 1000 architectures out of the 15K. This reveals another weakness, in order to rank the architectures, it does not have much choice but random sample a subset. This might work well to rank architecture with high performance in NATS-Bench, as you have roughly 1/15 probability to sample the highest performing model. However, if you generalize this method to a real-world search sapce, where you have billions of architectures, it might be hard to sample a good architecture when you random sample 1000 of them. If you sample a million for example, training to a small epoch will still be not feasible. This limits the generalization ability.

3. Experiments are too narrow, only showing on NATSBench is not enough IMHO, even for a workshop paper.


-- Justify --

I vote for rejection, please check the weakness for the reason.

---

### Decision · Program_Chairs · 2021-06-21

Accept (Poster)